# Nonlinear Optical Response of Reflective MXene Molybdenum Carbide Films as Saturable Absorbers

**DOI:** 10.3390/nano10122391

**Published:** 2020-11-30

**Authors:** Jiang Wang, Yonggang Wang, Sicong Liu, Guangying Li, Guodong Zhang, Guanghua Cheng

**Affiliations:** 1Center of Optical Imagery Analysis and Learning, Electronic Information College, Northwestern Polytechnical University, Xi’an 710072, China; wjiang@nwpu.edu.cn (J.W.); li_guang_ying@163.com (G.L.); guodongzhang@nwpu.edu.cn (G.Z.); 2School of Physics and Information Technology, Shaanxi Normal University, Xi’an 710119, China; chinawygxjw@snnu.edu.cn (Y.W.); liusicong8@163.com (S.L.)

**Keywords:** Molybdenum carbide, films, nonlinear optical materials, saturable absorbers, Q-switched

## Abstract

Molybdenum carbide (Mo_2_C) is a two-dimensional (2D) MXene material which makes it a promising photoelectric material. In this study, reflective type MXene Mo_2_C thin films were coated on a silver mirror by a magnetron sputtering method and were subsequently used in a passively Q-switched solid-state pulsed laser generator at the central wavelengths of 1.06 and 1.34 μm, respectively. The fabricated thin films of reflective type MXene Mo_2_C exhibited large modulation depth of 6.86% and 5.38% at the central wavelengths of 1064 and 1342 nm, respectively. By inserting the Mo_2_C saturable absorbers (SAs) into V-shaped Nd:YAG laser, short pulses were generated having a pulse duration, pulse energy, and average output power of 254 ns, 2.96 μJ, and 275 mW, respectively, at a wavelength of 1.06 μm. Similarly, shorter laser pulses were obtained in Nd:YVO_4_ laser at 1.34 μm. Our results illustrated potential of the 2D MXene Mo_2_C films for laser applications.

## 1. Introduction

Two-dimensional (2D) materials have been extensively studied due to their widespread applications in novel photonic devices [1,2] specifically in the development of ultrafast lasers [3,4,5,6]. In the field of ultrafast lasers, more nonlinear materials that combine the two advantages of broadband nonlinear modulation and ultrafast optical response have a new era of saturation absorbers (SAs). Semiconductor saturable absorption mirrors (SESAMs) are one of the most effective SAs. However, it has the disadvantage of complex manufacturing, narrow bands, and high prices. In recent years, SAs of 2D materials, such as grapheme [7,8,9,10], graphene oxide (GO) [11], topological insulators [12,13,14,15], transition-metal dichalcogenides [16,17,18,19], black phosphorus (BP) [20,21,22], and perovskites [23] have experienced rapid development in fabrication of passively Q-switched laser pulses. The SA of 2D materials can be used to modulate the pulse in a wide wavelength range of 1 to 3 μm, due to the broadband nonlinear modulation effect and easy integration into the passively Q-switched system.

Mxenes [24], which are transition metal carbides and nitrides, are the newly found 2D materials which have gained growing interest among the research community [25,26,27,28,29]. Furthermore, passively Q-Switched and mode-locked lasers with the MXene SA have attracted significant interest from scientists. In 2017, Jhon et al. reported a mode-locked Er-doped fiber laser with Ti_3_CNTx SAs operable at a wavelength of 1557 nm having a pulse width of 660 fs and an average output power of 0.05 mW [30]. Jiang et al. reported on an Yb-doped fiber lasers fabricated with Ti_3_C_2_Tx SAs. The shortest pulse duration and average output power are 480 ps and 9 mW, respectively [31]. In 2018, Feng et al. recorded a pulse duration of 359 ns, having an average output power of 94.8 mW for laser pulses from the Ti_3_C_2_Tx Q-Switched Nd:YAG laser [32]. In the same year, Wang et al. obtained Q-Switched pulses with a duration of 454 ns and an average output power only of 30 mW [33]. In 2019, Zu et al. used Ti_3_C_2_Tx in a Q-Switched Tm, Gd:CaF_2_ laser and the shortest pulse duration of 2.39 μs, having an average output power of 208 mW, was recorded [34]. Molybdenum carbide (Mo_2_C) is a new member of the MXenes family, which has been highlighted for its excellent conductivity, nontoxic synthesis processes, low cost, and high chemical stability [35,36]. In 2018, Tuo et al. reported on the mode-locked Er or Yb-doped fiber lasers, constructed using the Mo_2_C nanocrystal as SAs [37]. However, the reflective type Mo_2_C SAs has not been used in solid-state lasers until now.

Herein, we fabricated a Mo_2_C film coated silver mirror by magnetron sputtering. By inserting reflective type Mo_2_C SAs inside a Nd:YAG laser cavity, a stable Q-switched laser operation at a wavelength of 1064 nm with the maximum average output power of 275 mW corresponding to a pulse width of 254 ns and pulse energy of 2.96 μJ were obtained. In addition, the usage of reflective type 2D Mo_2_C SAs in Nd:YVO_4_ solid-state lasers operable at 1342 mm was demonstrated. The maximum average output power was 293 mW, the pulse width was 313 ns, and the pulse energy was 3.15 μJ.

## 2. Materials and Methods

### 2.1. Preparation of Mo_2_C SAs

The silver mirror was prepared by electron evaporation, wherein a layer of 180 nm silver was evaporated on a 1 mm thick quartz plate. An approximately 20 nm thick SiO_2_ layer was deposited on the silver film in order to protect the silver mirror. The Mo_2_C thin film was deposited on a silver mirror substrate by radio frequency magnetron sputtering deposition, as shown in the inset of Figure 1d. The vacuum level was set at 6 × 10^−4^ Pa and the finally obtained Mo_2_C sample was annealed at 600 °C.

### 2.2. Characterization of Mo_2_C SAs

The morphology of the Mo_2_C SAs was examined using the scanning electron microscopy (SEM, NovaNanoSEM Training-X50 series, FEI, Eindhoven, The Netherlands). The Mo_2_C SAs measurements were performed on a micro-Raman system (obtained using LabRam confocal Microprobe system, Horiba Jobin Yvon, Paris, France). An atomic force microscope (AFM Bruker Dimension ICON, Mannheim, Germany) was employed to observe the microstructures of Mo_2_C SAs nanostructures. The linear transmission spectra and nonlinear optical absorption of the samples were measured by the spectrophotometer (UV-Lambda 1050 PerkinElmer, Downers Grove, IL, USA) and homemade nanosecond pulsed laser (50 ns and 50 kHz) with the twin-detector measurement technique operated, respectively.

## 3. Results and Discussion

### 3.1. Characterization of Mo_2_C SAs Film

Figure 1a shows the 2D topographical images of the Mo_2_C films samples by AFM. Three different sections were chosen for the measurements to determine the thicknesses of the Mo_2_C thin films, as shown in Figure 1b. The thicknesses of the Mo_2_C films ranges from 12 to 16 nm were obtained. The micro structure of Mo_2_C films samples was determined by scanning electron microscopy, as shown in Figure 1c. Raman spectroscopy was used to investigate the composition of the Mo_2_C films samples. Figure 1d shows two noticeable peaks at 1344.5 (D band) and 1589.6 cm^−1^ (G band), which corresponds to the D and the G bands of carbon, respectively. In addition, the Raman spectra of the Mo_2_C films samples present one strong peak at 818.2 cm^−1^, which correspond to molybdenum oxide [36].

### 3.2. Nonlinear Optical Characteristics of Mo_2_C SAs Film

The reflectivity and nonlinear optical absorption of the sample were measured by a spectrophotometer and homemade pulsed Nd:YVO_4_ laser with balanced synchronous twin-detector technique [38], respectively. Figure 2a shows the reflectivity of Mo_2_C SAs was 52.6%@1064 nm and 88.6%@1342 nm. The laser source of central wavelength 1064 nm and 1342 nm were obtained by changing the end mirror (M1 and M2), respectively. We also used an actively Q-switched Nd:YVO4 solid-state laser with 50 ns pulse duration; and 5 KHz repetition rate; (A Actively Q-switched Nd:YVO4 solid-state laser with 50 ns pulse duration, and 5 KHz repetition rate, (AOM: Acousto Optical Modulator, VOA: Variable Optical Attenuator, FL: Focal Lens, PM: Power Meter, SAs: Mo2C SAs).), as shown in Figure 2b. The experimental data were fitted using the following equation [38]: *Τ*(*I*) = 1 − *α*_ns_exp(−*I*/*I*_sat_) − *Τ*_ns_
where *T*(*I*), *α*_ns_, *I*, *I*_sat_, and *T*_ns_ is the transmission, modulation depth, input intensity, saturation intensity and non-saturable loss, respectively. The modulation depth and saturation intensity of Mo_2_C SAs @1064 nm were 6.86% and 0.26 MW/cm^2^, respectively, as shown in Figure 2c. In addition, nonlinear optical transmission data at 1342 nm were shown in Figure 2d. The modulation depth and saturation intensity of Mo_2_C SAs were 5.38% and 0.62 MW/cm^2^, respectively. The damage threshold of Mo_2_C SAs film is estimated to be greater than 1 MW/cm^2^ (central wavelength 1064 nm).

### 3.3. Experimental Setup of Passively Q-Switched Nd:YAG Laser

Figure 3a is the scheme of Q-switched Nd:YAG solid-state laser. The diode-laser pump source is rated with power of 30 W at a wavelength of 808 nm. The gain medium is a Nd:YAG crystal doped at 1.2 at%. One end surfaces of the Nd:YAG crystal was coated with a high-transmission film at 808 nm and a high-reflection film at 1064 nm. As a reflection plane mirror of V-type laser, the combination of the SAs and reflector is beneficial to the simplification of the design of the laser. The laser V-folded cavity was designed to be 68 mm long. The mirror M1 (Radius of curvature is 100 mm) consists of an output coupler having a 10% transmission at 1064 nm obtained by adjusting the position of the SAs and the laser output from port P1 and port P2. Finally, the relationship between the average output power and the pump power was obtained, and the slope efficiency achieved was as high as 25.4%, as shown in Figure 3b.

Figure 4a depicts the pulse trains of Q-switched Nd:YAG solid-state lasers at the pump power of 4.2, 4.3, 4.7 and 4.8 W, respectively. Figure 4b shows the narrowest pulse duration of 254 ns at the pump power of 4.8 W. The variation of pulse duration and pulse repetition rate as a function of the incident pump power is shown in Figure 4c. The pulse duration decreased from 488 to 254 ns, and the pulse repetition rate increased from 49 to 93 kHz. The variation of the single pulse energy and the peak power of Q-switched pulse as a function of the incident pump power are shown in Figure 4d. When the pump power reaches 4.8 W, the maximum single pulse energy recorded was 2.96 μJ and the maximum peak power was 11.64 W. In general, the repetition rate of Q-switched pulses increases continuously with the augment of pump power, while the single pulse duration decreases. With the increase of pump power, the upper state population of SA level increases continuously, and the duration of pulse decreases. Continue to increaseAs the pump power continues to increase, the accumulation speed will slow down due to the upper state population soaredsoaring to over-saturation, leading to less a significant change of pulse duration. With pump power increases, the Q-switched operation could be still maintained, while the pulses have slight jitter. ItThis is mainly attributed to the degeneration of SA caused by laser induced heat accumulation [39].

### 3.4. Experimental Setup of Passively Q-Switched Nd:YVO_4_ Laser

Figure 5a depicts the schematic of Q-switched Nd:YVO_4_ solid-state laser. The gain medium is Nd:YVO_4_ crystal doped with 0.5 at% of Nd^3+^. The two-color mirror M1 was a plane mirror with anti-reflection and total reflection corresponding to wavelengths of 808 and 1342 nm, respectively. The mirror M2 (Radius of curvature is 100 mm) was an output coupler with a transmission of 5% at a wavelength of 1342 nm. The SAs was also used as a reflective plane mirror for a V-type laser. The laser V-folded cavity was designed to be 116 mm long. The crystal was pumped by diode laser (LD) emitting at a wavelength of 808 nm with a maximum output power of 30 W. The crystal temperature was maintained at 16 °C. By slightly adjusting the position of the SAs, the output of P1 and P2 of the two-channel Q-switched pulse laser was obtained. The relationship between absorbed pump powers has been shown in Figure 5b. As the pump power increased, the total average output power gradually increased. The highest slope efficiency was 18.8%.

Figure 6a depicts the pulse train diagram of Q-switched Nd:YVO_4_ solid-state laser at the pump power of 6.9, 7.1, 7.3, 7.5, and 7.6 W, respectively. The narrowest pulse duration recorded is 313 ns, which is shown in Figure 6b. The variation of the pulse duration and pulse repetition rate as a function of the incident pump power is shown in Figure 6c. The pulse duration decreases from 688 to 313 ns, while an increase in the pulse repetition rate increases from 55 to 93 kHz. The variation of the single pulse energy and peak power of the Q-switched pulse as a function of the pump power is shown in Figure 6d. When the pump power reaches 7.6 W, the maximum single pulse energy output is 3.15 μJ and the maximum peak power is 10.04 W with the pump power at 7.6 W. Table 1 summarizes the results obtained from passive Q-switched solid-state pulsed lasers. Compared to other 2D material absorbers, especially SAs based on material MXene [32,33,34], higher average power and narrower pulse width were obtained in this paper. The reason for this phenomenon may be the uniformity of the MXenes film. The smooth surface of the absorber reduces optical scattering, which can reduce the non-saturable losses of the absorber. The films prepared by magnetron sputtering deposition have better surface uniformity than those prepared by spin coating.

## 4. Conclusions

In summary, the reflective type MXene Mo_2_C thin films were fabricated on a silver mirror by magnetron sputtering method. The modulation depth of MXene Mo_2_C SAs at different bands is 6.86% (1064 nm) and 5.38% (1342 nm).By using the 2D Mo_2_C SAs, Nd:YAG, and Nd:YVO_4_ solid-state passively Q-switched pulsed lasers operating at the central wavelengths of 1.06 and 1.34 μm were obtained. The Nd:YAG laser generates a stable pulse train by changing the pump power from 4.2 to 4.8 W with a corresponding change in pulse width and average output power from 488 to 254 ns and 124 to 275 mW, respectively. The laser pulse duration of Nd:YVO_4_ is 313 ns, and the corresponding maximum average output power of 293 mW, pulse energy of 3.15 μJ and peak power of 10.07 W was obtained. Therefore, the reflective type 2D MXene Mo_2_C SAs has been proven to be a novel promising broadband saturable absorber for the fabrication of stable Q-switched pulse lasers.

## Figures and Tables

**Figure 1 nanomaterials-10-02391-f001:**
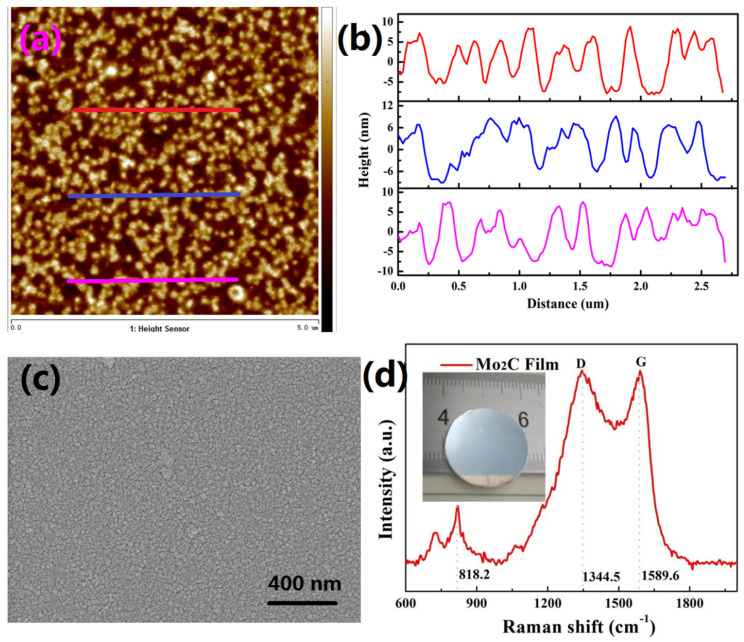
(**a**) Atomic force microscope (AFM) of Mo_2_C films are 2D topographical images, (**b**) their corresponding thickness profiles, (**c**) scanning electron microscopy (SEM) of Mo_2_C, and (**d**) Raman spectra of Mo_2_C film.

**Figure 2 nanomaterials-10-02391-f002:**
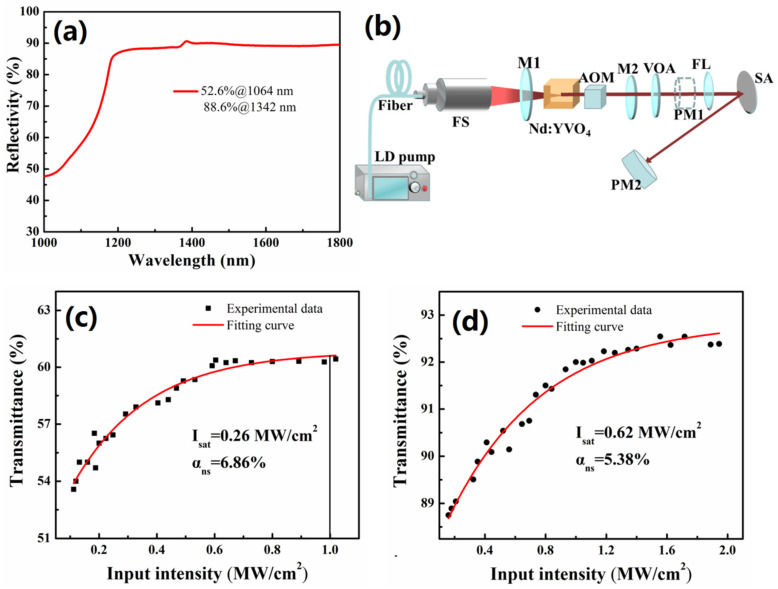
(**a**) Reflectivity spectrum, (**b**) schematic diagram of the experiment setup of nonlinear transmission measurement, nonlinear transmission of Mo_2_C SAs at (**c**) 1064 nm and (**d**) 1342 nm central wavelength.

**Figure 3 nanomaterials-10-02391-f003:**
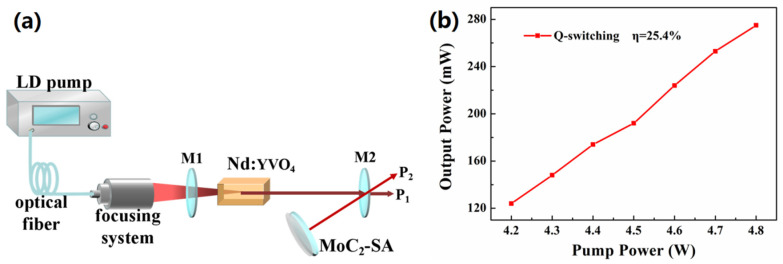
(**a**) Schematic depicting the Q-switched Nd:YAG solid-state lasers and (**b**) the relationship between average output power and input pump power.

**Figure 4 nanomaterials-10-02391-f004:**
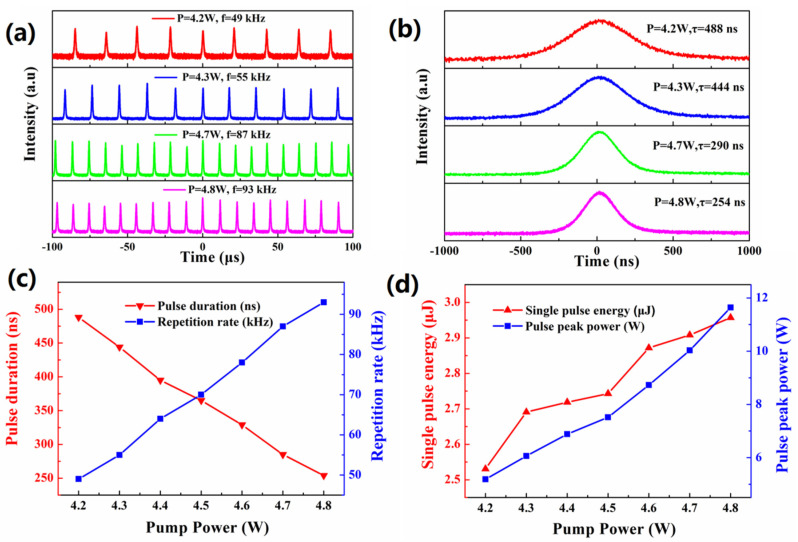
(**a**) Pulse trains and (**b**) Single pulse profiles of the Q-switched Nd:YAG laser; (**c**) Evolution of the repetition rate and pulse duration; (**d**) Single pulse energy and peak power curves with the pump power.

**Figure 5 nanomaterials-10-02391-f005:**
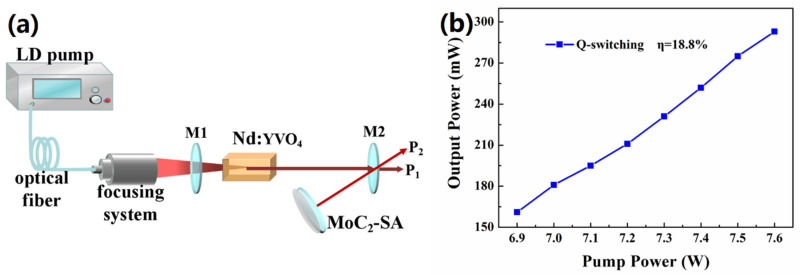
(**a**) Scheme of Q-switched Nd:YVO_4_ solid-state lasers and (**b**) the relationship between average output power and input pump power.

**Figure 6 nanomaterials-10-02391-f006:**
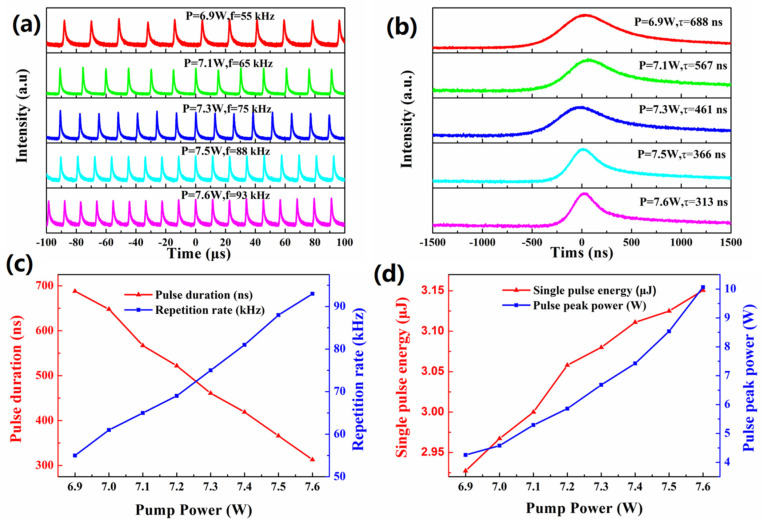
(**a**) Pulse trains and (**b**) Single pulse profiles of the Q-switched Nd:YVO_4_ laser; (**c**) Evolution of the repetition rate and pulse duration with the pump power; (**d**) Single pulse energy and peak power curves.

**Table 1 nanomaterials-10-02391-t001:** Q-switched s lid-state lasers based on 2D SAs.

SA Type	Laser Type	λ (μm)	τ (ns)	P (mW)	Ref.
Graphene	Nd:GdVO_4_	1.063	105	2300	[7]
Graphene	Nd:YAG	1.064	161	105	[8]
Graphene	Nd:GGG	1.331	556	690	[9]
Graphene	Nd:KLu(WO_4_)_2_	1.335	466	890	[10]
GO	Nd:YAG	1.064	156	1331	[11]
Bi_2_Te_3_	Yb:GAB	1.064	303	213	[12]
Bi_2_Te_3_	Yb:KGW	1.041	1600	439	[13]
Bi_2_Te_3_	Nd:YAG	1.3	673	326	[14]
Bi_2_Te_3_	Nd:LiYF_4_	1.313	443	198	[15]
MoS_2_	Yb:LGGG	1.025	182	600	[16]
MoS_2_	Nd:LuAG	1.342	188	180	[18]
MoS_2_	Nd:YAlO_3_	1.079	227	260	[19]
WS_2_	Nd:YVO_4_	1.064	2300	19.6	[38]
BP	Yb:CYA	1.046	620	37	[20]
BP	Nd:GGG	1.333	363	157	[21]
Perovskite	Nd:YVO_4_	1.342	222	236	[23]
Ti_3_C_2_Tx	Nd:YAG	1.064	359	94.8	[32]
Ti_3_C_2_Tx	Nd:YVO_4_	1.3	454	30	[33]
Ti_3_C_2_Tx	Tm, Gd:CaF_2_	1.929	2390	208	[34]
Mo_2_C	Nd:YAG	1.064	254	275	Our work
Mo_2_C	Nd:YVO_4_	1.342	313	293	Our work

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
