# Peer review of "Nonlinear Optical Response of Reflective MXene Molybdenum Carbide Films as Saturable Absorbers"

_nanomaterials, 2020, doi:10.3390/nano10122391_

Round 1

Reviewer 1 Report

In the manuscript with the title “Nonlinear Optical Response of reflective MXene Molybdenum carbide films as Saturable Absorbers” authors designed and generated passively Q-switched laser by using the Mo2C as a SAs for two different central wavelength such as 1064 nm and 1342 nm. Although authors successfully showed the passive mode-locking with Mo2C coating on the end mirror and get the pulsed laser with certain repetition rate depends on the pump power, there are a lot of points need to be improved in the manuscript where they are either misleading or unclear for the addressed audience. I suggest that this manuscript require major improvement before considering to be published in Nanomaterials. Please find some comments below;

  1. In the abstract the importance of the 2D materials should be emphasized as a SAs. Why we should care 2D SAs instead of traditional SAs?
  2. In the introduction line 31, the references are given in the same point like “….[7-23]..”, instead of this way, it could be nice it is separated for different materials and given the corresponding refs. after each materials like; “…such as graphene […], graphene oxide (GO) […],topological insulators […], transition-metal dichalcogenides […], black phosphorus (BP)[…], and perovskites[…]”
  3. Authors give good literature summary in the introduction about the where the transition metal carbides and nitrides have been used for the passively Q-switched systems. But the advantages/importance of the 2D is not highlighted. There is NO information and convincing reason for the importance which is mentioned by authors.
  4. There are some typing errors in the manuscript such as
    • line 57 it should be read ‘Wherein…’ instead of ‘wherein …’ and
    • line 75 there is extra F before ‘Figure 1 (b)’.

All manuscript needs to be double checked.

  1. Figure 1(b) looks like just a surface roughness profile not a thickness measurement. If the authors want to show the thickness with AFM measurements, they need to take a thickness profile from the substrate (quartz+SiO2 in their care) to coated (sputtered) region. They show in figure 1(b) the section which is completely coated with Mo2C. Also, it is not clear from that figure how they get the 15 nm thickness. Authors must explain thickness determination and maybe change the explanation in the caption of the corresponding figure.
  2. Figure 1 caption is not referring the figures properly. (b) and (c) should be swapped.
  3. In figure 3, there is no P1 and there are two P2, author should correct the typo there.
  4. In page 5, first paragraph authors explain their observation about the power dependent pulse duration and repetition rate. There is no explanation or discussion about the physics behind it. All manuscript is written just based on the observation. Regarding the physical meaning discussions, it is quite weak and needs to be improved significantly.
  5. Line 119, it is written “… duration grow shorter…” It is not clear it can be rewritten as “… duration decrease…”.
  6. Line 124, it is written that “When the pump power reaches 5.6 W,…” but in the plot this pump power does not matches according to output energy and max peak power. Authors must check the main text or the plot values, something is not consistent.
  7. It could be nice if the authors can write the pump power values for the figure 4 and 6 (a) and (b) in the figure caption. It will be easier to follow.
  8. Line 133, the curvature of the mirror is written with different way compared to line 109. It is better to keep consistency in the text to make the flow good.
  9. Line 150, same problem with comment 10. The pump power value doesn’t match with the plot x-axis.
  10. Figure 6 caption a and b is referring opposite, they need to be swapped.
  11. Regarding the Table 1, authors summarize the 2D material-based Q-switched solid state laser. That is good. But there is no comparison or discussion about them. Where is the Mo2C compared the others? What are the advantages? Why they picked the Mo2C? It should not be that there is a material let’s try this, authors must make it clear the motivation for their SAs. Moreover, regarding the conclusion, it is weak in terms of the discussion and explanation the physical mechanisms. They never talk about the damage threshold of the Mo2S. As we know, most of the 2D materials are good reflector in the IR and don’t get damage/burn since it doesn’t absorb in that spectral range. But these types of this discussions are missing in the manuscript. Conclusion should be rewritten and added the motivation of the Mo2C.
  12. Regarding the damage threshold and how they decide the pump power range, they need to add either text in the main manuscript or some experimental result.

Reviewer 2 Report

The authors prepared Mo2C MXene and studied nonlinear optical response as saturable absorbers in solid state laser . The paper is written well and I recommend this paper to be published in MDPI Nanomaterials after minor revision addressing following points:

  1. Molybdenum oxide is present probably as a result of Mo2C oxidation. Do you know the type of this oxide? Can you comment how the different types of Mo oxides (MoO2, MoO3, MoOx) would influence your nonlinear optical response?
  2. You mention that you prepared stable Q-switched pulse laser with Mo2C layer. Can you demonstrate that the Mo2C layer is not oxidized further and that the response is stable within days/weeks?

Round 2

Reviewer 1 Report

Thanks for the revisied version of the manuscript. Authors have changed and improved the manucript according the reviewers comments. I have no more comments and the changes are correct and enough to present their results. I suggest the publish the manuscript with revised version.